# Preclinical Evaluation of Recombinant Human IL15 Protein Fused with Albumin Binding Domain on Anti-PD-L1 Immunotherapy Efficiency and Anti-Tumor Immunity in Colon Cancer and Melanoma

**DOI:** 10.3390/cancers13081789

**Published:** 2021-04-09

**Authors:** Fei-Ting Hsu, Yu-Chang Liu, Chang-Liang Tsai, Po-Fu Yueh, Chih-Hsien Chang, Keng-Li Lan

**Affiliations:** 1Department of Biological Science and Technology, China Medical University, Taichung 406, Taiwan; sakiro920@mail.cmu.edu.tw (F.-T.H.); u409801001@ym.edu.tw (P.-F.Y.); 2Department of Radiation Oncology, Chang Bing Show Chwan Memorial Hospital, Lukang, Changhua 505, Taiwan; kevinyc.liu@gmail.com; 3Department of Radiation Oncology, Show Chwan Memorial Hospital, Changhua 500, Taiwan; 4Department of Medical Imaging and Radiological Sciences, Central Taiwan University of Science and Technology, Taichung 406, Taiwan; 5Department of Biomedical Imaging and Radiological Sciences, National Yang Ming Chiao Tung University, Taipei 112, Taiwan; amos.tcl@ym.edu.tw (C.-L.T.); chchang@iner.gov.tw (C.-H.C.); 6Institute of Traditional Medicine, School of Medicine, National Yang Ming Chiao Tung University, Taipei 112, Taiwan; 7Isotope Application Division, Institute of Nuclear Energy Research, Taoyuan 325, Taiwan; 8Department of Oncology, Taipei Veterans General Hospital, Taipei 112, Taiwan

**Keywords:** PD-L1, IL15, colon cancer, melanoma, tumor microenvironment

## Abstract

**Simple Summary:**

In this manuscript, we reported that a newly developed recombinant human IL15 fused with albumin binding domain (hIL15-ABD) showed superior biological half-life, pharmacokinetic and anti-tumor immunity than wild-type (WT) hIL15. Our hIL-15-ABD can effectively enhance anti-tumor efficacy of anti-PD-L1 on colon cancer and melanoma animal models. The anti-tumor potential of hIL-15-ABD was associated with tumor microenvironment (TME) regulation, including the activation of NK cells and CD8^+^ T cells, the reduction of immunosuppressive cells (MDSCs and Tregs) and the suppression of immunosuppressive factors (IDO, FOXP3 and VEGF). In conclusion, our new hIL15-ABD combined with anti-PD-L1 antibody increased the activity of anti-tumor effector cells involved in both innate and adaptive immunities, decreased the TME’s immunosuppressive cells, and showed greater anti-tumor effect than that of either monotherapy. We suggested hIL15-ABD as the potential complementary agent may effectively augment the therapeutic efficacy of anti-PD-L1 antibody in colon cancer and melanoma model.

**Abstract:**

Anti-PD-L1 antibody monotherapy shows limited efficacy in a significant proportion of the patients. A common explanation for the inefficacy is a lack of anti-tumor effector cells in the tumor microenvironment (TME). Recombinant human interleukin-15 (hIL15), a potent immune stimulant, has been investigated in clinical trial with encouraging results. However, hIL15 is constrained by the short half-life of hIL15 and a relatively unfavorable pharmacokinetics profile. We developed a recombinant fusion IL15 protein composed of human IL15 (hIL15) and albumin binding domain (hIL15-ABD) and explored the therapeutic efficacy and immune regulation of hIL-15, hIL15-ABD and/or combination with anti-PD-L1 on CT26 murine colon cancer (CC) and B16-F10 murine melanoma models. We demonstrated that hIL15-ABD has significant inhibitory effect on the CT26 and B16-F10 tumor growths as compared to hIL-15. hIL-15-ABD not only showed superior half-life and pharmacokinetics data than hIL-15, but also enhance anti-tumor efficacy of antibody against PD-L1 via suppressive effect on accumulation of Tregs and MDSCs and activation of NK and CD8+T cells. Immune suppressive factors including VEGF and IDO were also decreased by combination treatment. hIL15-ABD combined with anti-PD-L1 antibody increased the activity of anti-tumor effector cells involved in both innate and adaptive immunities, decreased the TME’s immunosuppressive cells, and showed greater anti-tumor effect than that of either monotherapy.

## 1. Introduction

Active immune system possesses fighting ability against tumor development and progression. An example of this is cytotoxic T lymphocytes (CD8^+^ T cells) and natural killer (NK) cells attacking tumor cells that can be elicited by antitumor immune signaling, resulting in tumor destruction [1,2]. However, tumors can escape immune surveillance through immunosuppressive tumor microenvironment (TME), restricting antitumor immunity. Immune checkpoints, which are composed of immunosuppressive molecule receptors or their ligands such as programed death receptor 1 (PD-1)/programed death ligand 1 (PD-L1) and cytotoxic T-lymphocyte-associated protein 4 (CTLA-4) modulate inactivation of CD8^+^ T and nature killer (NK) cells, resulting in tumor immune evasion in TME. Evasion of immune surveillance is conductive to tumor survival and progression [3,4,5].

Immunotherapy, an innovative therapeutic method that treats cancer by evoking anti-tumor immunity, is a promising strategy for treatment of solid tumors and hematologic malignancies [6,7,8]. Increased expression of PD-1/PD-L1 pathway is linked to T cell exhaustion and poor survival in multiple types of cancers. Blockade of PD-1/PD-L1 interaction with monoclonal antibodies reverses T cell exhaustion and prolongs survival benefit in patients with cancers such as melanoma, hepatocellular carcinoma (HCC), non-small-cell lung cancer (NSCLC), gastric, and urothelial cancers [7,9,10,11]. Furthermore, many preclinical and clinical studies have demonstrated that the therapeutic efficacy of PD-1/PD-L1 blocking antibodies can be enhanced with immunologic or non-immunologic agents [4].

Interleukin-15 (IL15), the immuno-oncology agent, potentiates antitumor immune via enhancement of CD8^+^ T and NK cells proliferation and cytotoxic activity. IL15 therapy has been demonstrated to attenuate tumor growth and improve survival rates in murine tumor models. IL15 is also recognized as a potential complementary agent to immunotherapy, effectively increasing anticancer immune response. The first clinical trial of hIL15 was conducted in patients with metastatic renal cell carcinoma and melanoma by daily intravenous administration for 12 consecutive days of recombinant hIL15 expressed by *Escherichia coli* [12,13,14]. Although some encouraging clinical results were observed, the bioactivity of IL15 is limited due to short in vivo half-life. N-803, formerly ALT-803, composed of N72D IL15 mutant, sushi domain of IL15Rα, and Fc domain of human IgG1, has been demonstrated to have longer serum half-life and more potent stimulatory effect on NK cells and T-lymphocytes than that of WT hIL15 [15,16]. N-803 has been shown to boost antitumor response of anti-PD-L1 antibody in triple negative breast and colon cancers in vivo and its combination with anti-PD-1 monoclonal antibody, Nivolumab, has been verified in safety to treat refractory metastatic non-small cell lung cancer patients with observed tumor responses [17].

We have generated a recombinant fusion protein (hIL15-ABD), which is composed of human IL15, albumin binding domain (ABD), and hexahistidine tag (his6). hIL15-ABD could be expressed by *E. coli* and refolded into active fusion protein, which simultaneously binds to albumin and stimulate CTLL-2 proliferation as well as downstream signaling pathway evidenced by enhanced STAT5 phosphorylation. Fusion of hIL15 with ABD greatly enhanced pharmacokinetic parameters, including half-life, Cmax and area under curve (AUC) as compared with those of hIL15 in experimental mice. hIL15-ABD also displayed significant inhibitory effect on the tumor growths of CT26 murine colon cancer (CC) and B16-F10 murine melanoma models. Moreover, combination of hIL15-ABD with a rat antibody against murine PD-L1 antibody, 10F.9G2, demonstrated greater anti-tumor effect than that of either monotherapy, by enhancing the activity of anti-tumor effector cells associated with both innate and adaptive immunities as well as decreasing the TME’s immunosuppressive cells.

## 2. Materials and Methods

### 2.1. Reagents and Antibodies

FITC Rat Anti-Mouse CD3 (#561798), PerCP-Cy™5.5 Rat Anti-Mouse CD4 (#561115), FITC Rat Anti-Mouse CD8a (#561966), PE Rat Anti-Mouse CD25 (#561065), FITC Rat Anti-CD11b (#561688), PE Rat Anti-Mouse CD49b (DX5, #561066), PerCP-Cy™5.5 Rat Anti-Mouse CD335 (#560800), Alexa Fluor^®^ 488 Rat anti-Mouse Foxp3 (#560407), PE Rat Anti-Mouse Ly-6G and Ly-6C (Gr-1, #561084), PerCP-Cy™5.5 Mouse Anti-Mouse NK-1.1 (#561111) and Foxp3 Fixation/Permeabilization Buffer Set (#560409) were all purchased from BD Pharmingen™ (BD Biosciences, San Diego, CA, USA). Cleaved-caspase-3 (E-AB-30004, Elabscience Biotechnology Inc, Houston, TX, USA), BAX (#50599-2-lg, Proteintech Inc., Rosemont, IL, USA), Ki-67 (#E-AB-2202, Elabscience Biotechnology Inc.), granzyme B, Indoleamine 2,3-dioxygenase (IDO), Forkhead box protein P3 (FOXP3), CD49b and Interferon-gama (IFN-γ Rat Anti-Mouse PD-L1 (10F.9G2, Bioxcell, Lebann, NH, USA) antibodies were all purchased from different companies as listed.

### 2.2. In Vitro Characterization for CTLL2 Stimulation and Albumin Binding of hIL15-ABD

CTLL-2 cells in logarithmic phase were harvested, washed and resuspended in RPMI-1640 medium supplemented with 10% FBS at a concentration of 8 × 10^3^ cells per well of 96-well plate. hIL15-ABD in 1 μL of various refolding buffers was incubated with 100 μL of CTLL-2 cell culture of 96-well at 37 °C for 2 days followed by addition of 20 μL MTS reagent (CellTiter 96^®^ AQueous Non-Radioactive Cell Proliferation Assay, Promega, Madison, WI, USA) according to manufacturer’s instructions. The viable CTLL-2 cells were measured at 490 nm on a TECAN Sunrise™ multichannel microtiter plate reader. For STAT5 phosphorylation assay, CTLL2 cells were treated with increasing concentrations of either hIL15 or hIL15-ABD for 20 min followed by fixation with formaldehyde (2% *v*/*v* final concentration) for 15 min at room temperature. Fixed cells were then spun down (×500 g), and cell pellets were permeabilized with ice cold 100% methanol and incubated on ice for 20 min. Cells were rehydrated by washing twice with 250 μL of PBS in the presence of 0.8% BSA. Cells were incubated with antibody against phosphorylated STAT5 for 16 h at 4°, washed twice with PBS with 0.5% BSA, and lastly treated with FITC-conjugated anti-rabbit IgG for 60 min at room temperature in the dark. The positive events were detected with a BD FACSCalibur flow cytometer and analyzed with CellQuest Pro and Cytexpert software (Beckman Coulter). To examine the ability of hIL15-ABD for binding to albumin, human or murine albumin was diluted in PBS and immobilized on an ELISA plate by incubation at 4 °C overnight. Albumin coated well were blocked with 300 μL 3% milk for 2 h at room temperature, followed by washing the plate three times with wash buffer (0.05% Tween-20 in PBS). Various concentrations of hIL15 or hIL15-ABD were incubated with immobilized human or murine albumin at room temperature for one hour, followed by washing the plate three times with wash buffer. The *in vitro* binding of his6-tagged hIL15 and hIL15-ABD with human albumin was detected using an HRP-tagged, anti-his6 antibody and developed by the addition of the HRP substrate (100 μL/well), 3,3′,5,5′-tetramethylbenzidine (TMB). The peroxidase reaction was stopped 20 min after the addition of 0.5 M H_2_SO_4_ (50 μL/well), and the absorbance was measured at 450 nm with a multichannel microtiter plate reader.

### 2.3. Expression, Refolding and Purification of hIL15 and hIL15-ABD

BL21 (DE3) *E. coli* strain transformed with plasmids encoding hIL15 or hIL15-ABD was cultured with terrific broth (TB)/ampicillin (100 μg/mL) and grew at 37 °C in a shaker at 250 rpm. When reaching logarithmic phase with OD_600_ at 0.7, the *E. coli* culture was treated with increasing concentration of IPTG ranging from 0, 0.01, 0.05, 0.1, 0.5, to 1 mM. Additionally, the cultures were further incubated at either 30 or 37 °C overnight. The induced cultures were centrifuged at 6000 rpm for 15 min, and supernatant was removed followed by re-suspending the pellet in PBS. The pellet was subjected to continuous high-pressure cell disrupter twice at pressure of 28 kpsi followed by centrifugation at 4500 rpm for 15 min for removal of supernatant. The pellets were washed six-time with either 200 mL of H_2_O or Tris-HCl buffer in the presence or absence of 1%SDS, and 0.5 M NaCl. Each washing steps were followed by centrifugation at 6000 rpm for 15 min. The inclusion body pellets were solubilized with by 8 M urea buffer containing 25 mM imidazole and loaded onto Ni-NTA resin column. Elution of immobilized protein was conducted by stair-wise increase of imidazole from 75, 300, to 500 mM of imidazole in PBS buffer (pH 7.5). Fractions containing hIL15 or hIL15-ABD were pooled and dilution refolded to a concentration of 0.1 mg/mL by slow dripping into refolding buffer matrix with various combinations of Triton (0.05%), EDTA (2 mM), NaCl (250 mM), GSH (1 mM)/GSSG (0.1 mM), and L-Arginine (0.4 M). The refolding process lasted for 24 h at 4°. After examining activities of hIL15ABD and hIL15 for albumin binding and STAT5 phosphorylation, buffer 26 containing NaCl (250 mM), GSH (1 mM) and GSSG (0.1 mM) in Tris-HCl buffer (50 mM, pH 8.5) was selected for later large-scale protein refolding.

### 2.4. In Vitro Characterization for CTLL2 Stimulation and Albumin Binding of hIL15-ABD

The ability of hIL15 and hIL15-ABD for T-cell activation was examined for either proliferation of or phosphorylation of STAT5 in CTLL2 cells using MTS assay and flow cytometry, respectively. The detail procedure was described in material and methods section.

### 2.5. Pharmacokinetics

Female Balb/c mice (*n* = 3, 6 weeks, 20 g) received an intraperitoneal injection of either hIL15 (1 µg) or equimolar hIL15-ABD (1.5 µg) in 300 µL of PBS. For mice injected with hIL15, in time intervals of 15 min, 30 min, 45 min, 1 h, 2 h, 4 h, 6 h, whereas for mice treated with hIL15-ABD, in time intervals of 15 min, 30 min, 45 min, 1 h, 2 h, 4 h, 8 h, 24 h, 36 h, and 52 h, blood samples were withdrawn from the tail and placed on ice. Serum samples were obtained by centrifuging clotted blood at 800 g for 10 min at 4 °C. Serum concentrations of hIL15 and hIL15-ABD were determined by ELISA specific for human hIL15 (DY247-05, R&DSystem, Minneapolis, MN, USA). Pharmacokinetic parameters were determined using the Phoenix^®^ WinNonlin software version 7.0 (Certara USA Inc., Princeton, NJ, USA). Noncompartmental analysis (extravascular input) was used with the log/linear trapezoidal rule. Parameters, including terminal half-life (T_1/2λz_), Tmax, Cmax and area under the curve (AUC) were determined. Pharmacokinetic parameters associated with the terminal phase were calculated using the last four measured time points to estimate the terminal half-life.

### 2.6. Cell Culture

CT26 mouse colon cancer (BCRC #60447) cell line and B16-F10 (BCRC #60031) mouse melanoma were purchased from the Bioresource Collection and Research Center (Hsinchu, Taiwan). Cells were cultured in Roswell Park Memorial Institute (RPMI) 1640 medium and Dulbecco’s Modified Eagle medium (DMEM) supplemented with 10% heat-inactivated FBS, 2mMl-glutamine, 100units/mL penicillin and 100μg/mL streptomycin in a humidity atmosphere containing 5% CO_2_ and at 37 °C. Cells culture related reagents and medium were all purchased from Gibco BRL, Grand Island, NY, USA.

### 2.7. Transfection and Stable Clone Selection

The vector containing CMV-luciferase2 vector (pGL4.50[luc2/CMV]) (Promega, Madison, WI, USA) and transfection reagent (Polyplus transfection, France) were prepared in advance. B16-F10 cells were seeded in 6 cm plate one day before transfection. Cells density is around 70% during transfection procedure. The jetPEI^TM^ reagent (10 µL) dissolved in 250 µL of NaCl buffer was then added into DNA buffer (5 μg plasmids with 250 µL of NaCl buffer), the mixture was then incubated at 25 °C for 25 min. The mixture was finally added to the B16-F10 cells for an incubation period of 1 day. Luc2 expression cells were selected by hygromycin B 200 μg/mL for another two weeks and named as B16-F10/*luc2* cells [18].

### 2.8. Immune Cells (CD8^+^T Cells and NK Cells) Validation

The CD8^+^T and NK cell percentages and functions were used to evaluate immune activation status. CD8^+^ T cell percentage and function were identified by CD8, IL-2, and IFN-γ markers in tumor-draining lymph node (TDLN) and spleen. Intracellular staining was performed with Fixation/Permeabilization kit following the manufacturer’s protocol. In addition, NK cells on two different strains of animal models were also identified by various markers, CD3^-^/CD49b^+^/CD335^+^ on BALB/c and CD3^-^/CD49b^+^/NK1.1^+^ on C57BL/6, respectively [19]. The percentages of these cell types were acquired by NovoExpress^®^ flow cytometry (Agilent, Santa Clara, CA, USA) and data was analyzed by FlowJo software (BD Pharmingen™).

### 2.9. Immune Suppressive Cells (Treg Cells and MDSCs) Validation

The percentages of regulatory T cells (Tregs) and myeloid-derived suppressor cells (MDSCs) cells were used to evaluate immunosuppressive function. Immune suppressive cells isolated from tumor-draining lymph node (TDLN), spleen [20], and bone marrow (BM), were stained with anti-FOXP3-Alexa Fluor 488/CD4-PerCP-Cy™5.5/CD25-PE antibodies using a Mouse Treg Flow Kit according to manufacturer’s protocol. CD11b-FITC/Gr-1-PE antibodies were used to detecting Tregs and MDSCs [21], respectively. The percentages of these cell types were acquired by NovoExpress^®^ flow cytometry and data were analyzed by FlowJo software (BD Pharmingen™).

### 2.10. Animal Experiments

The animal experiments were performed in accordance with the protocols approved by the Animal Care and Use Committee at China Medical University (approval number: CMU IACUC-2019-208). Six-week-old male BALB/c and C57BL/6 mice were purchased from the National Laboratory Animal Center and housed in a pathogen-free animal facility. The establishment of animal model was described in material and methods section. All experiment was repeated at least twice (*n* = 6).

### 2.11. Animal Treatment Procedure

The animals were anaesthetized with 1–2% isofluorane during surgery and imaging. The animals were fed sterilized mouse chow and water. Five million of CT26 or B16F10/luc2 cells were administered to mice (20–25 g) by subcutaneous injection on right thigh. The body weight and tumor volume were measured 3 times per week. Tumor volume was calculated by following formula: volume = length × width^2^ × 0.523. The animals were separated into various groups and administered with 100 µL of indicated treatment by i.p.: control (DMSO 0.1%), hIL15 (5 µg/injection), hIL15-ABD (5 µg/injection), or 10F.9G2 alone (anti-PD-L1, 100 µg/injection), co-treatments of hIL15-ABD and 10F.9G2. The drugs for animal treatment were dissolved in 100 µL H_2_O with 0.1% DMSO.

### 2.12. Enzyme-Linked Immunosorbent Assay (ELISA)

Secreted IL15, and VEGF were collected from serum and assayed by ELISA. All the procedures followed commercially provided protocol. IL15, and VEGF ELISA kits were all purchased from Elabscience (Houston, TX, USA). ELISA readings were determined by OD scanning at 450 nm using SpectraMax iD3 microplate reader from (Molecular Devices, Downingtown, PA, USA).

### 2.13. Bioluminescence Imaging (BLI)

Mice bearing B16-F10/*luc2* tumors of each group (*n* = 6) were intraperitoneally injected with 200 μL of 150 mg/kg D-luciferin in PBS before anesthetization with 1–2% isoflurane 10 min prior to imaging. Mice were then set onto the imaging platform and continuously exposed to 1–2% isoflurane throughout the time. The luc2 signal from tumor region was collected by BLI using IVIS50 Imaging System (Xenogen) once per week. The photons emitted from the tumor were assayed using IVIS50 Imaging System with an acquisition time of 1 min. Regions of interest (ROIs) were drawn around the tumor and quantified with the Living Image software as photons/s/cm^2^/sr.

### 2.14. Immunohistochemistry (IHC)

Formalin-fixed and paraffin-embedded tissues from mice were subjected to IHC staining. In brief, sections of paraffin-embedded tumor tissue on slides obtained from each group was deparaffinized in xylene, rehydrated with decreasing concentrations of ethanol (100%, 70%, 30%, 0%), and then incubated in 3% H_2_O_2_ for 10 min. After washing, the slides were blocked with 5% normal goat serum for 5 min in a tight container, followed by incubation with different primary antibodies in a dilution of 1:100–500 at 4 °C overnight. Finally, slides were counterstained with hematoxylin. At least three slides from each group were studied. Slides were photographed at 200 × magnifications by Nikon ECLIPSE Ti-U microscope and quantified by ImageJ software (National Institutes of Health, Bethesda, MD, USA).

### 2.15. Statistical Analysis

Statistical analysis was performed utilizing excel 2017 software (Microsoft, Redmond, WA, USA) and GraphPad *Prism* 8.0 (GraphPad Software, Inc., San Diego, CA, USA). Values were expressed as means ±SD. Comparison of means between several groups were performed by one-way analysis of variance (ANOVA) and independent-test was used to compare between two groups. Tukey’s test was used to compare all groups as post-hoc test. Values were considered statistically significant at *p* ≤ 0.05.

## 3. Results

### 3.1. Expression and Purification of Active hIL15-ABD

hIL15-ABD expression by transformed *E. coli*, BL21(DE3), were initiated when OD of culture reached 0.7, in the presence of increasing concentration of IPTG, ranging from 0, 0.05, 0.1, 0.3, 0.5, to 1 mM) at either 30 or 37 °C and 200 rpm. SDS-PAGE (upper panel, Figure 1A) and Western blot (lower panel, Figure 1A) analysis displayed comparable expressions of hIL15-ABD (21.0 kDa) induced by all the IPTG concentrations and two temperatures tested. Large scale protein expression was initiated by addition of 0.1 mM IPTG to 1 L of transformed *E. coli* culture when OD600 value reached 0.7 and kept in rotating shaker at 37 °C and 200 rpm (Figure 1). The majority of expressed hexahistidine-tagged hIL15-ABD was in the inclusion bodies, which were washed and dissolved in an 8 M urea denaturing buffer (Figure 1B) before being loaded onto a Ni-NTA resin column. The immobilized hexahistidine-tagged hIL15-ABD was eluted sequentially using 75, 300, to 500 mM of imidazole in PBS buffer (Figure 1C). The most significant portion of hexahistidine-tagged hIL15-ABD was eluted with 300 mM as displayed by SDS-PAGE (Figure 1D) and Western blotting using anti-hexahistidine antibody (Figure 1E). The purified denatured hIL15-ABD was investigated for optimal refolding condition using buffer matrix as listed in Table 1, and the bioactivities of the refolded hIL15-ABD were examined for binding to human albumin (Figure 1F) as well as stimulation CTLL-2 proliferation (Figure 1G). It turns out that refolding of ABD moiety of hIL15-ABD was quite robust in most of the buffers examined, whereas there is more significant difference in stimulatory effects of hIL15-ABD refolded in individual buffers. We selected buffer 26 composed of NaCl (250 mM), GSH (1 mM) and GSSG (0.1 mM) in Tris-HCl buffer (50 mM, pH 8.5) for later large-scale protein refolding (Figure 2A). We were able to obtain approximately 60 mg of recombinant hIL15-ABD with purity higher than 90% per liter of TB culture in shake flasks IPTG-induced *E. coli.* The bioactivities of the refolded hIL15-ABD were examined for binding to either human or murine albumin as well as stimulation of STAT5 phosphorylation in CTLL-2 cells. hIL15-ABD demonstrates similar binding affinity for both human and murine albumin with Kd values of 3.0 and 2.8 nM, respectively (Figure 2B), whereas it displayed comparable stimulatory effect on STAT5 phosphorylation with that of hIL15 positive control with EC50 of 0.17 and 0.10 nM, respectively (Figure 2C).

### 3.2. Pharmacokinetics Studies

The serum concentration-time curves from derived from Balb/c mice intraperitoneally injected with 1 μg hIL15 and equimolar of 1.5 μg hIL15-ABD are shown in Figure 2D. The pharmacokinetic parameters are summarized in Table 2. The maximum serum concentrations (Cmax) and times to reach Cmax (Tmax) were determined as 13.59 ng/mL at 0.75 h for hIL15 and 51.28 ng/mL at 4 h for hIL15ABD, respectively. The terminal half-lives (T_1/2λz_) of hIL15 and hIL15-ABD were 0.88 h and 23.37 h after injection, respectively. The results showed that T_1/2λz_ of hIL15-ABD was 26-fold longer than that of hIL15 in serum, which confirmed that the long circulation of the hIL15-ABD has been achieved. The AUC (0→∞) of hIL15 and hIL15-ABD were 18.8 ng/mL×h and 1602.4 ng/mL×h, respectively. The AUC (0→∞) of hIL15-ABD in serum was 180-fold larger than that of hIL15.

### 3.3. hIL15-ABD Showed Superior Tumor Growth Inhibition and Positive Regulation of Immune Response on CC Model

After confirming that hIL15-ABD has more than 20- and 80-fold increase of biological half-life and AUC, respectively, compared to those of hIL15 (Table 2), we further validated the treatment efficacy of both on colon cancer-bearing animal model (Figure 3A). In light of the superior pharmacokinetic profiles of hIL15-ABD and to demonstrate the potent in vivo anticancer effect of hIL15-ABD, we used 5 μg for each injection, which represent 0.36 and 0.24 nanomole of hIL15 and hIL15-ABD, respectively, instead of using equimolar proteins. As shown in Figure 3B and C, hIL15-ABD displayed better tumor growth inhibition ability as compared to hIL15 past day 9 post-treatment. Additionally, hIL15-ABD showed potential to suppress the accumulation of MDSCs, immunosuppressive cells, in spleen and bone marrow (Figure 3D). Percentage of CD11b^+^/Gr-1^+^ cells that are recognized as MDSCs were effectively decreased in hIL15-ABD treated group (Figure 3E,F). Percentage of another group of immunosuppressive cells, regulatory T cells (Tregs), from TDLN and spleen was also identified by flow cytometry after treatment (Figure 3G). Number of CD4^+^/CD25^+^/FOXP3^+^ cells was more effectively reduced around one of two by hIL15-ABD as compared to hIL15 (Figure 3H,I). The Treg population was reduced by more than a half in hIL15-ABD-treated group compared to non-treatment group, and also had significantly larder reduction compared to hIL15-treated group. Furthermore, we observed that hIL15-ABD may also increase the population of CD8^+^ T cells in TDLN and spleen (Figure 3J). Two times more percentage of CD8^+^ T cells was detected in hIL15-ABD group compared to non-treatment group (Figure 3K,L). Other than induction of adaptive immunity, NK cells, which plays role in innate immunity, was also effectively triggered in hIL15-ABD-treated group (Figure 3M). CD3^-^/CD49b^+^, CD3^-^/CD335^+^ and CD3^-^/CD49b^+^/CD335^+^ cells population were all significantly elevated in hIL15-ABD-treated group compared to hIL15-treated group (Figure 3N). Decrease in mice body weight was only observed in hIL15-treated group (Figure 3O), indicating the possibility of toxicity caused by prolonged treatment with unmodified form of hIL15. In sum, hIL15-ABD not only demonstrates better tumor inhibition, but also provides a positive microenvironment for cells involved in innate and adaptive immunities to function.

### 3.4. hIL15-ABD Enhanced Tumor Inhibition Capacity and Triggered Apoptosis Effect of Anti-PD-L1 Therapy on Both CC and Melanoma Models

Though hIL15-ABD monotherapy demonstrated tumor inhibition potential, the inhibition ability remained limited. Therefore, we further validated whether hIL15-ABD may positively augment the function of checkpoint inhibitor-related therapy. In Figure 4A, we show the effects of hIL15-ABD and anti-PD-L1 anti-body (10F.9G2) monotherapies as well as their combined therapy effect. Combined therapy not only showed superior tumor growth inhibition in colon cancer (CC) (Figure 4B), but also melanoma bearing animal model (Figure 4D). Tumors isolated from combined therapy groups in CC and melanoma models on day 21 displayed significant tumor shrinking effect as compared to those isolated from monotherapy groups (Figure 4C,E). In addition, tumor weight of combined therapy groups also showed more significant decreases compared to either hIL15-ABD or anti-PD-L1 anti-body monotherapy groups (Figure 4F,G). Luc2 signal emitted from melanoma (B16-F10/*luc2*) was recognized as amount of living cells within tumor region that also presented the minimal signal intensity in combined therapy group (Figure 4H). Quantification result from BLI (Figure 4I) was corresponded to tumor volume, and the combined therapy group was found to exhibit superior tumor growth inhibition compared to the monotherapy groups. No obvious body weight loss of each treatment procedure was found in both CC and melanoma models (Figure 4J,K). Ki-67, a cell proliferation marker, was showed to be effectually suppressed by combination therapy (Figure 4L,M). Finally, we measured BAX and cleaved caspase-3 protein expression levels in CC and melanoma (Figure 4N). BAX and cleaved caspase-3 stain signals were markedly increased in combination therapy group (Figure 4O,P). Taken together, these results demonstrate that hIL15-ABD can successfully enhance tumor inhibition ability of anti-PD-L1 by disrupting proliferation effect and induction of apoptosis signaling.

### 3.5. hIL15-ABD Strengthened Anti-PD-L1-Induced Function of CD8^+^ T Cells on Both CC and Melanoma Models

To further investigate the effect of hIL15-ABD and anti-PD-L1 combination on tumor microenvironment, we determined the function of CD8^+^ T cells by observing activation of intracellular IFN-γ and IL-2. As shown in Figure 5A, CD8^+^ cells with the expression of IFN-γ were increased to 50% in combination treatment in TDLN. Both CC and melanoma displayed an increasing percentage of IFN-γ in CD8^+^ cells from TDLN after combination therapy (Figure 5B). The expression level of IFN-γ in CD8^+^ Tcells from SP has showed similar elevations, especially in combination therapy group (Figure 5C,D). Furthermore, we found that IL-2 activation in CD8^+^ T cells from TDLN (Figure 5E,F) and SP (Figure 5G,H) were both effectually increased in the hIL15-ABD + anti-PD-L1 combined therapy group as compared to monotherapy groups. Lastly, we performed IHC staining on CC and melanoma tumor to validate granzyme B and CD8 protein expression after therapy (Figure 5I). The results show that not only CD8, but also granzyme B, key indicators of cytotoxic T cells (CD8^+^ T), were raised by combination therapy (Figure 5J,K). Higher activation levels of IFN-γ and IL-2 in CD8^+^ T cells from TDLN and SP in CC and melanoma models were also found in the combination treatment group relative to those of the monotherapy groups.

### 3.6. hIL15-ABD Increased Anti-PD-L1 Antibody Induced Accumulation and Activation of NK Cells in Both CC and Melanoma Models

To identify whether combining hIL15-ABD with anti-PD-L1 promote the function of NK cells, we measured NK cell population and activity in the spleen after treatment. Results from Figure 6A indicate that CD3^-^/CD49b^+^, CD3^-^/CD335^+^, CD3^-^/NK1.1^+^, CD3^-^/NK1.1^+^/CD335^+^ and CD3^-^/CD49b^+^/CD335^+^ cells were all dramatically increased after combination therapy. Based on different species of animal, we separated NK cells according to their specific markers as indicated in Figure 6B,C. Highest amount of CD3^-^/NK1.1^+^/CD335^+^ and CD3^-^/CD49b^+^/CD335^+^ triple positive cells were found in the combined therapy group. Next, we identified whether these NK cells possessed function by detecting intracellular IFN-γ. The activation of IFN-γ was observably increased in CD3^-^/NK1.1^+^ and CD3^-^/CD49b^+^ cells from the combined therapy group (Figure 6D,E). Furthermore, we also investigated the expression levels of CD49b and IFN-γ proteins expression on tumor tissue from CC and melanoma models by IHC staining (Figure 6F). As shown in Figure 6G,H, the protein expression levels of both CD49b and IFN-γ were increased in treated groups. Finally, we checked VEGF (Figure 6I) secretion level in mouse serum to demonstrate the decreasing of immunosuppressive factor after combination therapy. Most importantly, the level of IL15 secretion was also effectively triggered by hIL15-ABD combined with anti-PD-L1 (Figure 6J). These results support the hypothesis that hIL15-ABD combined with anti-PD-L1 may develop a positive regulation of immune response for defending against tumor.

### 3.7. hIL15-ABD Combined Anti-PD-L1 Antibody Diminished the Accumulation of Immunosuppressive Cells in Both CC and Melanoma Models

Next, we determined whether hIL15-ABD promotes anti-tumor capacity of anti-PD-L1 by reducing accumulation of Tregs and MDSCs. Flow cytometry from mice TDLN showed that CD4^+^/CD25^+^/FOXP3^+^ triple positive cells amount was significantly reduced in combination therapy group (Figure 7A). The amount of Tregs was decreased by around 5–10 fold as compared to non-treated control (Figure 7B). At the same time, the percentage of Tregs decreased the most in the combination treatment group (Figure 7C,D). Moreover, we also detected the population of MDSCs within BM and SP of CC and melanoma mice by flow cytometry. The obtained results of flow cytometry from BM indicated the effective diminishment of CD11b^+^/Gr-1^+^ MDSCs in combination treatment group (Figure 7E,F). The percentage of CD11b^+^/Gr-1^+^ MDSCs within CC and melanoma mice SP was also decreased after combination therapy (Figure 7G,H). Subsequently, we validated the protein expression level of FOXP3 and IDO in mice tumor by IHC staining (Figure 7I). FOXP3 and IDO are known to be important immunosuppressive factors that allow the tumor to escape immunosurveillance. As indicated in Figure 7J,K, proteins expression levels of FOXP3 and IDO in combination therapy group were decreased to 10–30% of that in non-treated control. Our results illustrate that combination of hIL15-ABD and anti-PD-L1 may develop an environment that inhibits the tumor’s ability to escape from immune surveillance.

## 4. Discussion

In the first human clinical trial, hIL15, as a wild-type (WT) recombinant protein was administrated for 12 consecutive days to patients with metastatic melanoma and renal cell carcinoma [12]. Dose-limiting toxicities of WT hIL15, included grade 3 hypotension, thrombocytopenia, and elevated values of ALT and AST and 0.3 μg/kg per day was determined as the maximum tolerable dose. Although greatly altered homeostasis of lymphocyte subsets, such as NK cells and memory CD8 T cells, as well as anticancer efficacy observed in the first in-human trial of recombinant WT hIL15, it becomes evident that alternative dosing strategies is needed to enhance efficacy while reducing toxicity. Non-human primate pharmacokinetic study verified that constant administration regimens of recombinant IL15 through either continuous intravenous infusion or subcutaneous injection achieve remarkable immune stimulation in the absence of obvious toxicity, indicating potentially better clinical result than the previous bolus intravenous regimen [22]. Clinical trial of recombinant hIL15 administrated subcutaneously daily (Monday through Friday) for two weeks was conducted in patients with refractory solid tumor cancers. This dosing regimen resulted in markedly enhanced circulating CD56^bright^ NK and CD8^+^ T cells as well as an encouraging safety profile [23].

Although hIL15 displayed encouraging results in early clinical trials, its short half-life suggests potential improvement in anticancer efficacy through engineering hIL15 with prolonged half-life. N-803, the novel hIL15 superagonist complex, comprises N72D mutant IL15 and IL15Rα-IgG Fc fusion protein and displays enhanced affinity for IL-2Rβ and prolonged half-life. It is under multiple clinical trials, including advanced melanoma, renal cell, non-small cell lung, head and neck, hematologic malignancies who relapse after allogeneic hematopoietic cell transplantation and showing encouraging results [16,17,24]. N-803 has been shown to exhibit greater anti-CC activity compared to hIL15 in CT26 bearing model [25]. In this study, CT26 bearing model was also used to evaluate differences in therapeutic efficacy and anticancer immune response between hIL15 and hIL15-ABD treatments. Our results demonstrate that hIL15-ABD group has higher tumor growth inhibition capability and anticancer immunity than hIL15 group (Figure 3). Immunosuppressive cells such as Tregs and MDSCs restrain antitumor immunity through the downregulation of effector T cells and NK cells [26,27,28]. The increased abundance of Treg or MDSCs in peripheral blood and tumor are associated with poor prognosis in different types of cancer [29,30,31]. Although the relationship between Treg population and prognosis in patients with colorectal cancer remains uncertain [32,33], depletion of Tregs and MDSCs has been indicated to promote anticancer immunity in colorectal cancer [34,35]. In our results, we present that hIL15-ABD not only significantly increased percentage of CD8^+^ T and NK cells (Figure 3J–O), but also effectively reduced population of Tregs and MDSCs compared to hIL-15 treatment (Figure 3D–I).

Anti-PD-L1 therapies have been shown promising results as a member of an increasing number of immunotherapies against cancer [36]. However, despite its potential, anti-PD-L1 antibody has failed to elicit objective response in a majority of patients treated [37]. A common explanation for the lack of response is the lack of anti-tumor effector cells in the TME. Both tumor cells and MDSCs express PD-L1, which binds to PD-1 on T cells and causes T cell exhaustion as well as conversion of T helper type 1 (Th1) cells to Tregs. The combination of N-803 and anti-PD-L1 therapy reduced numbers of Tregs and MDSCs in lung [38,39]. Having verified that hIL15-ABD is superior to hIL15 in inhibiting tumor growth and regulating anti-cancer immunity, we investigated the anticancer efficacy and immune response induction of hIL15-ABD combined with anti-PD-L1 in both CC and melanoma models. Our results indicate an obvious enhancement of tumor growth inhibition in CT26 or B16-F10 bearing mice after hIL15-ABD and anti-PD-L1 combined therapy (Figure 4B–G). Furthermore, the combination group had significantly smaller population of Tregs (within TDLN and SP, Figure 7A–D) and MDSCs (within BM and SP, Figure 7E–H) compared to hIL15-ABD or anti-PD-L1 therapy monotherapies.

Both CD8^+^ T and NK cells are critical executors that mediate tumor cell apoptosis through secretion of granzyme-B and IFN-γ in immunotherapy modulating tumor regression. In addition to hIL15, anti-PD-1/-L1 therapy has also been indicated to enhance anti-tumor efficacy of CD8^+^ T and NK cells [4,9,40,41,42]. The combination of N-803 and anti-PD-L1 therapy significantly induced the activated CD8^+^ T cell phenotype compared to N-803 or anti-PD-L1 monotherapies in murine breast cancer models [43]. The increased number and function of CD8^+^ T or NK cells were linked to favorable prognosis in patients with colorectal cancer or melanoma [44,45,46]. Therefore, it is worthwhile to investigate whether hIL15-ABD promotes anti-PD-L1 therapy-elicited activity and percentage of CD8^+^ T and NK cells in CT26 or B16-F10 bearing mice. In our results, the combination of hIL15-ABD and anti-PD-L1 monotherapy effectively increased percentage of CD8^+^IFN-γ or CD8^+^IL-2 cells in spleen and TDLN compared to hIL15-ABD or anti-PD-L1 therapy (Figure 5A–H). Human IL15-ABD also significantly promoted anti-PD-L1 therapy-induced accumulation and function of NK cells in spleen and TDLN (Figure 6A–H).

Granzyme-B, the granule protease secreted by NK and CD8^+^ T cells, induces apoptosis through BAX/BAK-mediated mitochondrial apoptotic pathway [47]. The increased level of granzyme-B in serum or tumor was correlated with favorable outcomes in patients with colorectal cancer or NSCLC [48,49]. In our results indicated the combination group presented significantly higher expression of granzyme-B and apoptotic proteins (BAX and cleaved-caspase-3) in CT-26 or B16-F10 tumor tissues compared to hIL15-ABD or anti-PD-L1 therapy (Figure 5I–K and Figure 4N–P). VEGF, the major angiogenic mediator, contributes to tumor growth and metastasis through promoting new vessel formation. VEGF participates in regulation of Tregs and MDSCs leading to restriction of anti-tumor immunity. The high level of serum VEGF was correlated with poor overall survival of melanoma patients treated with the immune checkpoint inhibitor [50]. Indoleamine 2,3-dioxygenase (IDO), immunosuppressive protein, attenuates anti-tumor function of T cells by regulating the conversion of tryptophan to kynurenine [51]. The decreased expression of IDO was associated with better prognosis in patients with colorectal cancer or melanoma [52]. Our results demonstrated expression of IDO and VEGF was obviously reduced by hIL15-ABD, anti-PD-L1, or combination therapy (Figure 6I and Figure 7I–K). The combination group had lower expression of IDO or VEGF compared to hIL15-ABD or anti-PD-L1 monotherapy in CT26 or B16-F10 bearing mice.

In light of the great potential of IL15 as one of the critical weapons in the arsenal of anticancer immunotherapy, many related therapeutics, ranging from the wild-type IL15, IL15 superagonist (N-803) to PD-L1–targeting IL15 (KD033 [53] and N-809 [54]), are actively being developed in either preclinical or clinical settings. In our study, hIL15-ABD displays a much-extended half-life and superior inhibitory effect on CT26 and B16 growth in experimental mice than WT hIL15, which has shown encouraging results in early human trial [12]. hIL15-ABD could be easily purified and refolded into active form with yields of approximately 60 and 300 mg/L of transformed *E. coli* TB cultures in shake flask and fermenter, respectively, indicating a relatively lower production cost than those of modified hIL15, such as N-803, KD033 and N-809, expressed by mammalian cells. Intriguingly, while displaying anticancer effect as a monotherapy (Figure 3B), hIL15 treated CT26-bearing mice showed statistically significant body weight loss as compared with those treated with vehicle and hIL15-ABD (Figure 3O), suggesting a better therapeutic window of hIL15-ABD comparing with hIL15. Given that albumin-based carriers for anticancer therapeutics has shown promising results in both preclinical and clinical studies not only through half-life prolongation but also enhanced tumor localization [55], it is of great interest to investigate whether albumin associated hIL15-ABD will obtain a more favorable biodistribution profile, thereby increasing anticancer effects while reducing toxicity to normal organs.

## 5. Conclusions

In conclusion, for the first time, we presented the hIL15-ABD, the novel recombinant IL15 protein, was superior to in induction of tumor regression and antitumor immunity. hIL15-ABD may suppress the accumulation of MDSCs and Treg at the site of the tumor. In addition, hIL15-ABD can also promote the activity of IL-2 and IFN-γ in CD8^+^ T cells or NK cells, supporting more effective anti-tumor activity by effector cells. Importantly, hIL15-ABD can trigger innate immunity by enhancement of NK cells toxicity effect. Furthermore, the combination of hIL15-ABD and anti-PD-L1 therapy significantly inhibited tumor growth and promoted anti-tumor immune response compared to either monotherapy in mouse models of CC or melanoma. We demonstrated enhancement of CD8^+^ T and NK cells accumulation and cytotoxic function and reduction of Tregs and MDSCs population are associated with antitumor properties of hIL15-ABD combined with anti-PD-L1 therapy in CC or melanoma. We suggested the combination of hIL15-ABD and anti-PD-L1 therapy as potential immune therapy may offers therapeutic activity for treatment of CC or melanoma.

## Figures and Tables

**Figure 1 cancers-13-01789-f001:**
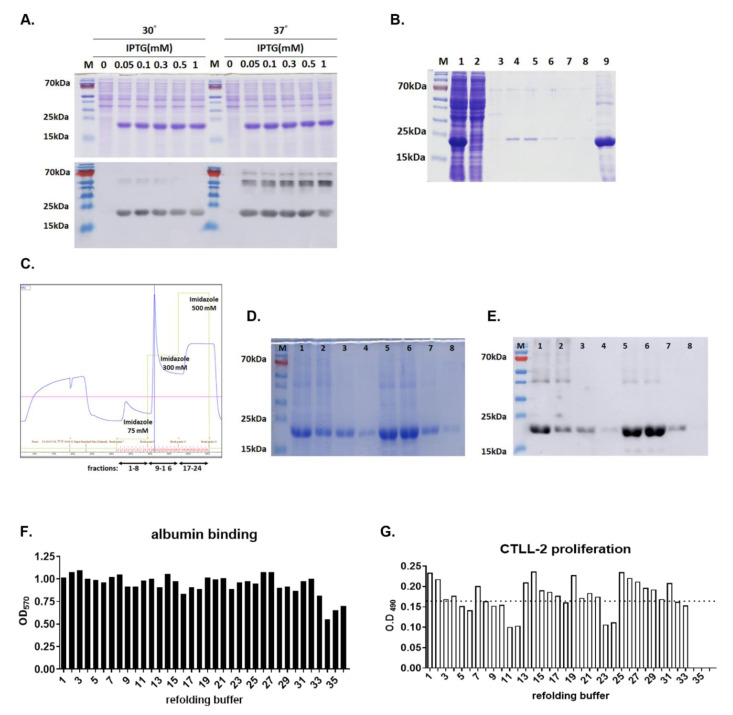
hIL15-ABD expression, refolding and purification. (**A**) SDS-PAGE (upper panel) and Western blot (lower panel) analysis of hIL15-ABD expression induced with increasing concentrations of IPTG, ranging from 0, 0.05, 0.1, 0.3, 0.5, to 1 mM in transformed *E. coli*, BL21(DE3) at either 30 or 37 °C. (**B**) SDS-PAGE analysis of fractions from transformed *E. coli* lysates (lane 1); supernatant of the lysates after centrifugation at 10,000 rpm for 20 min (lane 2), supernatant after washing with H_2_O (lane 3); supernatant after washing with 20 mM Tris-HCl (lane 4); supernatant after washing with 50 mM Tris-HCl buffer containing 2 mM EDTA and 0.1% SDS (lane 5); supernatant after washing with 50 mM Tris-HCl, 150 mM NaCl and 2 mM EDTA (lane 6); supernatant after washing with H_2_O (lane 7); supernatant after washing with H_2_O (lane 8); denatured inclusion body in 8 M urea (lane 9). Lane 1 to 8 each are loaded protein equal to 75 μL and lane 9 equal to 37.5 μL of culture medium. (**C**) Purification of solubilized inclusion body from *E. coli* expressing hIL15-ABD through Ni-column. The blue curve indicates the absorption at 280 nm in mAU, whereas the green line represents the concentration of imidazole. (**D**) SDS-PAGE and (**E**) western blot analysis of elution fractions number 3–6 (lane 1–4), 10, 11, and 12 (lane 5–7), and 21 (lane 9). (**F**, **G**) The purified denatured hIL15-ABD was refolded with buffer matrix listed in Table 1 and resulted L15-ABD is examined for (**F**) human albumin binding and (**G**) stimulation of CTLL-2 proliferation. The dot line indicates the average OD490 values representing the extents of viable CTLL-2 in 96-well plates cultured with refolded hIL5-ABD.

**Figure 2 cancers-13-01789-f002:**
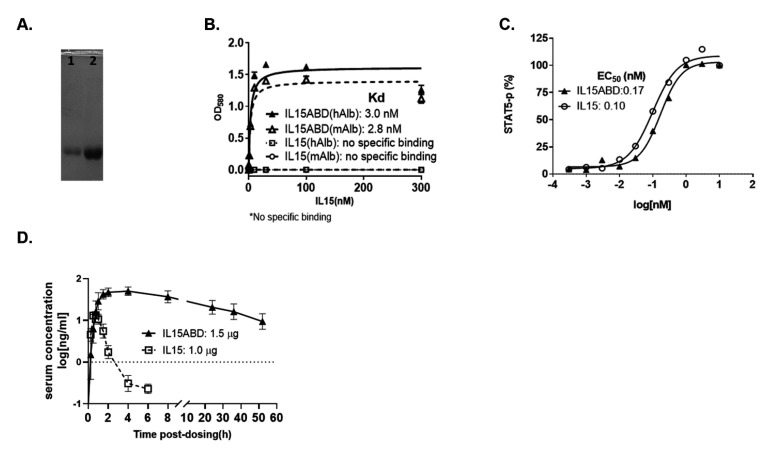
*In vitro* characterizations and pharmacokinetics of hIL15-ABD purified in large scale. (**A**) Lane 1 of the SDS-PAGE indicates the purified hIL1-ABD in refolding buffer number 26 (NaCl (250 mM), GSH (1 mM) and GSSG (0.1 mM) in Tris-HCl buffer (50 mM, pH 8.5)) and lane 2 represents hIL15-ABD after being condensed following the process of refolding. (**B**) Refolded purified hIL-15-ABD displays similar affinity for binding to both human and murine albumin, whereas there is no measurable specific binding to albumin by hIL15. (**C**) hIL15 and IL15-ABD demonstrate comparable EC50 values of stimulation of STAT-5 phosphorylation in CTLL-2 cells, which are 0.10 and 0.17 nM, respectively. (**D**) Serum concentration-time curves of hIL15 in Balb/c mice following single intraperitoneal injection of hIL15-ABD and hIL15 at 1.5 and 1.0 μg/mouse. Data are presented as mean ± SD. N = 3 at each time point.

**Figure 3 cancers-13-01789-f003:**
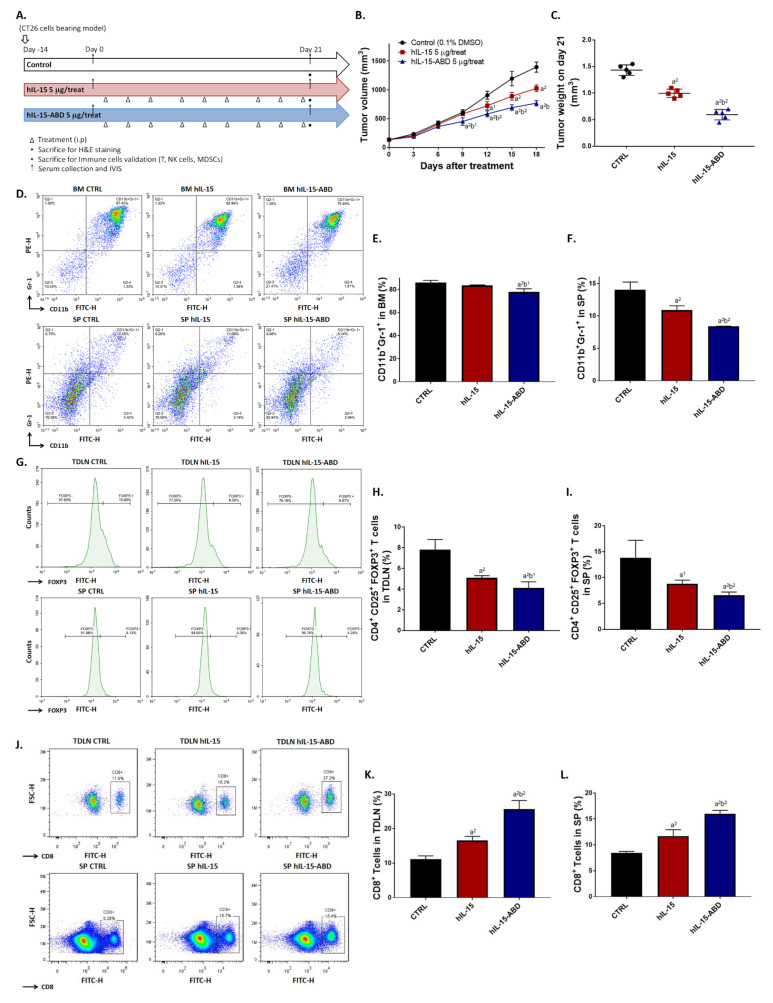
hIL15-ABD induced the accumulation of CD8^+^ T cells and NK cells, but diminished Tregs and MDSCs, resulting in colon cancer growth inhibition. (**A**) Animal flow chart of different treatment materials is displayed. (**B**) Tumor volume is recorded every 3 days and (**C**) tumor weight is weighed after isolation from mice on day 21. (**D**) Flow cytometry pattern of CD11b+/Gr-1+ MDSCs isolated from BM and SP. Percentage of CD11b^+^/Gr-1^+^ MDSCs from (**E**) BM and (**F**) SP are gated and quantified by FlowJo software. (**G**) Flow cytometry pattern of CD4+/CD25+/FOXP+ Tregs isolated from TDLN and SP. Percentage of CD4^+^/CD25^+^/FOXP3^+^ Tregs from (**H**) TDLN and (**I**) SP. (**J**) Flow cytometry pattern of CD8^+^ T cells isolated from TDLN and SP. Percentage of CD8^+^ T cells from (**K**) TDLN and (**L**) SP. (**M**) Flow cytometry pattern of CD3^-^/CD49b^+^/CD335^+^ NK cells isolated from TDLN and SP. Percentage of CD3^-^/CD49b^+^/CD335^+^ NK cells from (**N**) SP. (**O**) Mice body weight are measured 3 time per week. [BM = bone marrow, TDLN = tumor-draining lymph node and SP = spleen] (a^1^
*p* < 0.05, a^2^
*p* < 0.01 vs. CTRL; b^1^
*p* < 0.05, b^2^
*p* < 0.01 vs. hIL15).

**Figure 4 cancers-13-01789-f004:**
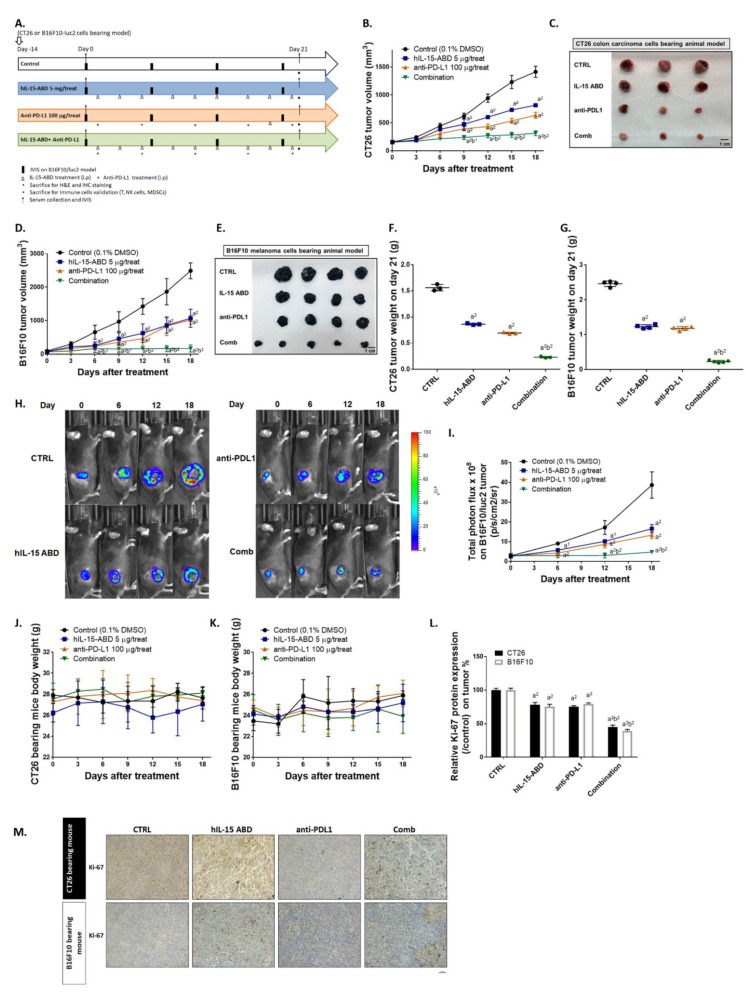
hIL15-ABD facilitated anti-tumor efficacy of anti-PD-L1 antibody via enhancing apoptosis mechanism. (**A**) Animal flow chart of hIL15-ABD, anti-PD-L1 and combination treatment is presented. (**B**,**C**) Colon cancer (CC) tumor growth from day 0–18 and tumor photographed on day 21 are displayed. (**D**,**E**) Melanoma tumor growth from day 0–18 and tumor photographed on day 21 are displayed. Tumor weight from (**F**) CC and (**G**) melanoma on day 18 are summarized. (**H**) BLI and (**I**) quantification results from B16-F10/*luc2* bearing mice are presented. Mice body weight from (**J**) CC (**K**) melanoma model is recorded every 3 days during therapy. (**M**,**N**) IHC staining images and (**L**,**O**,**P**) relative proteins quantification level on CC and melanoma are presented. (a^1^
*p* < 0.05, a^2^
*p* < 0.01 vs. CTRL; b^1^
*p* < 0.05, b^2^
*p* < 0.01 vs. hIL15-ABD and anti-PD-L1; scale bar = 100 μm).

**Figure 5 cancers-13-01789-f005:**
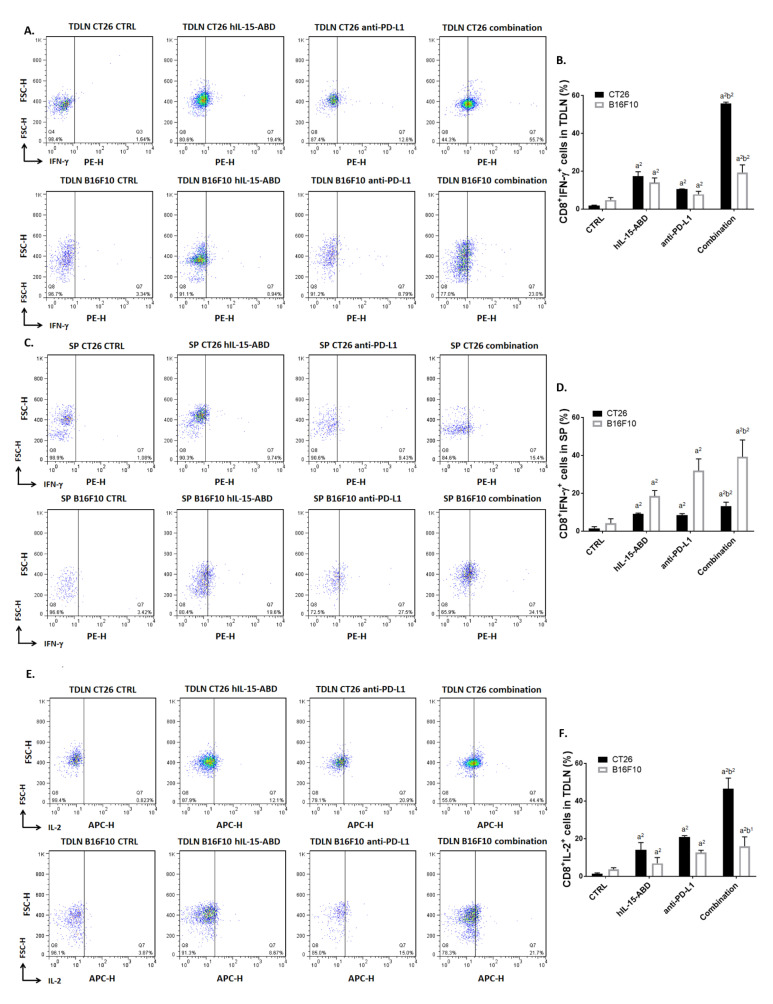
hIP-15-ABD offer a reinforce role of increasing anti-PD-L1 antibody induced CD8+ T cells activation. (**A**) Flow cytometry pattern and (**B**) quantification results of CD8+/IFN-γ^+^ cells from TDLN. (**C**) Flow cytometry pattern and (**D**) quantification results of CD8^+^/IFN-γ^+^ cells from SP. (**E**) Flow cytometry pattern and (**F**) quantification results of CD8^+^/IL-2^+^ cells from TDLN. (**G**) Flow cytometry pattern and (**H**) quantification results of CD8^+^/IL-2^+^ cells from SP. (**I**) Granzyme B and CD8 immunohistochemistry (IHC) staining images and relative proteins quantification level of (**J**) CC and (**K**) melanoma are displayed. (a^1^
*p* < 0.05, a^2^
*p* < 0.01 vs. CTRL; b^1^
*p* < 0.05, b^2^
*p* < 0.01 vs. hIL15-ABD and anti-PD-L1; scale bar = 100 μm).

**Figure 6 cancers-13-01789-f006:**
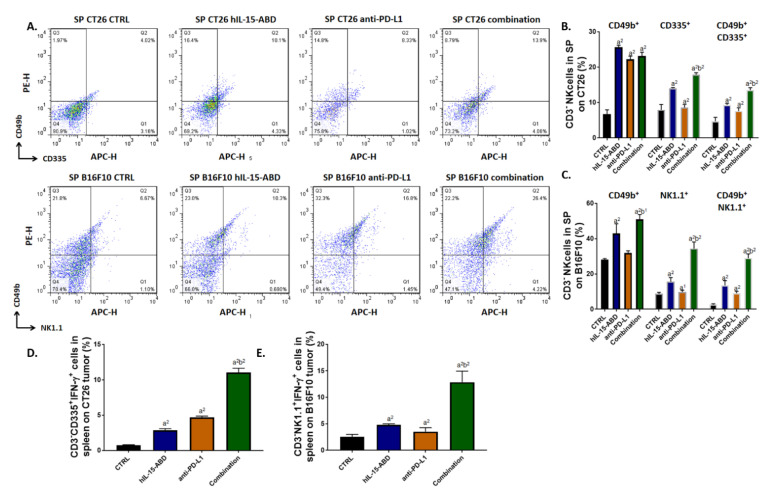
hIP-15-ABD combined anti-PD-L1 antibody effectively trigger the accumulation and function of NK cells. (**A**) Flow cytometry pattern and (**B**) quantification results of CD3^-^/CD49b^+^, CD3^-^/CD335^+^ and CD3^-^/CD49b^+^/CD335^+^ NK cells from SP on CC bearing BALB/c animal model. (**C**) Quantification results of CD3^-^/CD49b^+^, CD3^-^/NK1.1^+^ and CD3^-^/CD49b^+^/NK1.1^+^ NK cells from SP on melanoma bearing C57BL/6 animal model. (**D**,**E**) CD3^-^/CD335^+^/IFN-γ^+^ and CD3-/NK1.1^+^/IFN-γ^+^ NK cells from SP on CC and melanoma model is displayed. (**F**) CD49b and IFN-γ IHC staining images and relative proteins quantification level on (**G**) CC and (**H**) melanoma are shown. Expression level of secreted (**I)** VEGF and (**J**) IL15 are shown as quantification results. (a^1^
*p* < 0.05, a^2^
*p* < 0.01 vs. CTRL; b^1^
*p* < 0.05, b^2^
*p* < 0.01 vs. hIL15-ABD and anti-PD-L1; scale bar = 100 μm).

**Figure 7 cancers-13-01789-f007:**
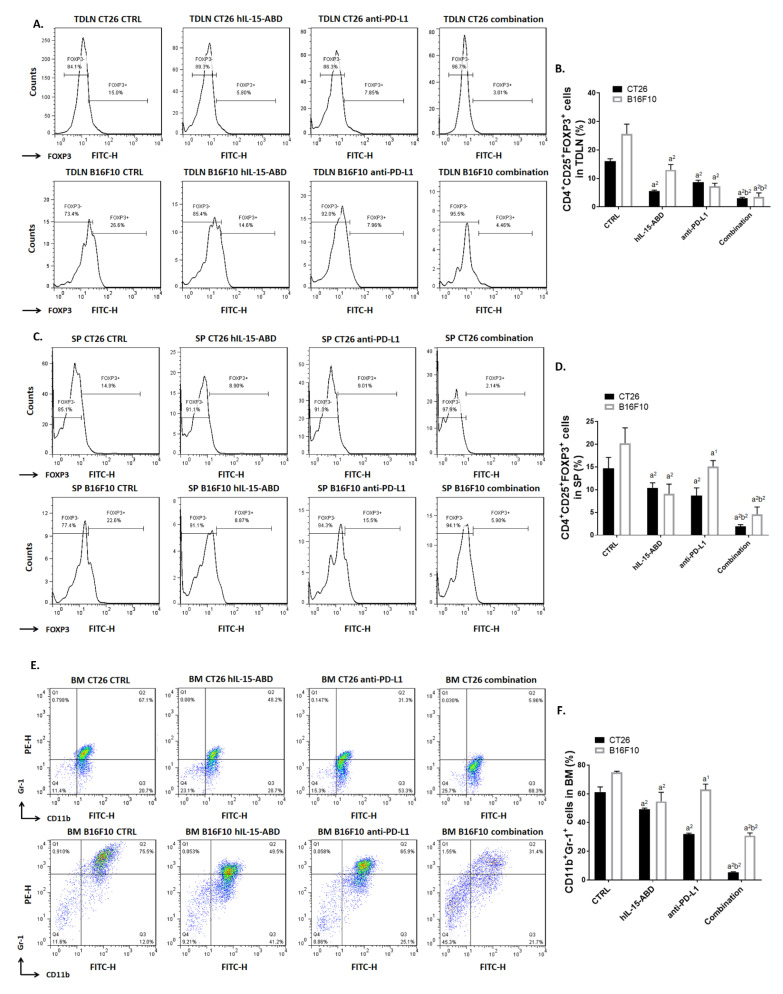
hIP-15-ABD combined anti-PD-L1 antibody successfully suppress the accumulation of immunosuppressive cells. (**A**,**C**) Flow cytometry pattern and (**B**,**D**) quantification results of CD3^-^/CD49b^+^, CD4^+^/CD25^+^/FOXP3^+^ Tregs from TDLN and SP, respectively. (**E**,**G**) Flow cytometry pattern and (**F**,**H**) quantification results of CD11b^+^/Gr-1^+^ MDSCs from BM and SP, respectively. (**I**) FOXP3 and IDO IHC staining images and (**J**,**K**) relative quantification of CC and melanoma are presented. (a^1^
*p* < 0.05, a^2^
*p* < 0.01 vs. CTRL; b^1^
*p* < 0.05, b^2^
*p* < 0.01 vs. hIL15-ABD and anti-PD-L1; scale bar = 100 μm).

**Table 1 cancers-13-01789-t001:** Combinations of buffer used for protein refolding.

	Triton (0.05%)		EDTA (2 mM)		NaCl (250 mM)	
	^1^ GSH	^2^ Arginine	^1^ GSH	^2^ Arginine	^1^ GSH	^2^ Arginine
50 mM Tris-HCl pH6.5	1	7	13	19	25	31
50 mM Tris-HCl pH8.5	2	8	14	20	26	32
50 mM Tris-HCl 1 M Urea pH6.5	3	9	15	21	27	33
50 mM Tris-HCl 1 M Urea pH8.5	4	10	16	22	28	34
50 mM Tris-HCl 1 M GdnHCl pH6.5	5	11	17	23	29	35
50 mM Tris-HCl 1 M GdnHCl pH8.5	6	12	18	24	30	36

^1^ 1 mM GSH/0.1 mM GSSG; ^2^ 0.4 M L-Arginine.

**Table 2 cancers-13-01789-t002:** Pharmarcokinetic parameters of hIL-15 and hIL-15-ABD after intraperitoneal injection in BALB/c mice.

Parameter	Unit	hIL-15	hIL-15-ABD
		Value	Value
T1/2λz	h	0.88	23.37
Tmax	h	0.75	4.00
Cmax	ng/mL	13.59	51.28
AUC(0→∞)	ng/mL×h	18.90	1602.4

Calculated with WinNonlin 7.0 for a noncompartmental model.

## Data Availability

The data generated and analyzed will be made available from the corresponding author on reasonable request.

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
