# Peer review of "Preclinical Evaluation of Recombinant Human IL15 Protein Fused with Albumin Binding Domain on Anti-PD-L1 Immunotherapy Efficiency and Anti-Tumor Immunity in Colon Cancer and Melanoma"

_cancers, 2021, doi:10.3390/cancers13081789_

Round 1

Reviewer 1 Report

This is an innovative study that adds to the increasing literature on IL-15 and the various modified forms of IL-15.

A brief description of the other forms some of which are in clinical trials would add to its interest.

Where do they think their construct fits in regard to these other forms?

They also do not mention the role of IL-15 in T resident memory cells which some regard as critical against cancers like melanoma.

Author Response

Cancers

RESVISED RESPONSES TO REVIEWERS

Manuscript Number: cancers-1111292

Manuscript Title: Preclinical evaluation of recombinant human IL15 protein fused with albumin binding domain on anti-PDL1 immunotherapy efficiency and anti-tumor immunity in colon cancer and melanoma

Authors: Fei-Ting Hsu, Yu-Chang Liu, Chang Liang Tsai, Po-Fu Yueh, Chih-Hsien Chang, Keng-Li Lan

Dear Reviewer

First of all, we would like to thanks for your valuable advices. We revised the main text according to your valuable comments, and all the revised words and sentences were marked as red color. Original comments from reviewer were written in black color, and the revised information was written as blue color as follow.

Major points:

This is an innovative study that adds to the increasing literature on IL-15 and the various modified forms of IL-15.

Reply: Thanks for reviewer comment.

A brief description of the other forms some of which are in clinical trials would add to its interest.

Reply: Thanks for reviewer valuable comment. As shown below, three paragraphs regarding the indications and early findings of the clinical trials of WT IL15 and N-803 (formerly ALT-803) have been added into the manuscript.

  1. The first clinical trial of IL15 was conducted in patients with metastatic renal cell carcinoma and melanoma by daily intravenous administration for 12 consecutive days of recombinant IL15 expressed by E. coli1 (page 2).
  2. In the first human clinical trial, IL15, as a wild-type (WT) recombinant protein was administrated for 12 consecutive days to patients with metastatic melanoma and renal cell carcinoma1. Dose-limiting toxicities of WT IL15, included grade 3 hypotension, thrombocytopenia, and elevated values of ALT and AST and 0.3 mg/kg per day was determined as the maximum tolerable dose. Although greatly altered homeostasis of lymphocyte subsets, such as NK cells CD8 memory T cells, as well as anticancer efficacy observed in the first in-human trial of recombinant WT IL15, it becomes evident that alternative dosing strategies is needed to enhance efficacy while reducing toxicity. Nonhuman primate pharmacokinetic study verified that constant administration regimens of recombinant IL15 through either continuous intravenous infusion or subcutaneous injection achieve remarkable immune stimulation in the absence of obvious toxicity, indicating potentially better clinical result than the previous bolus intravenous regimen. Clinical trial of recombinant IL15 administrated subcutaneously daily (Monday through Friday) for two weeks was conducted in patients with refractory solid tumor cancers. This dosing regimen resulted in markedly enhanced circulating CD56bright NK and CD8 T cells as well as an encouraging safety profile (page 20).
  3. Although IL15 displayed encouraging results in early clinical trials, its short half-life suggests potential improvement in anticancer efficacy through engineering IL15 with prolonged half-life. N-803, the novel IL15 superagonist complex, comprises N72D mutant IL15 and IL15Rα-IgG Fc fusion protein and displays enhanced affinity for IL-2Rβ and prolonged half-life. It is under multiple clinical trials, including advanced melanoma, renal cell, non-small cell lung, head and neck, hematologic malignancies who relapse after allogeneic hematopoietic cell transplantation and showing encouraging results (page 20).

Where do they think their construct fits in regard to these other forms?

Reply: In addition to the two IL15 related therapeutics, WT IL15 and N-803, currently in clinical trials, we added more information on two PD-L1 targeting IL15, KD033 and N-809 in the manuscript. Additionally, we outlined the potential advantages of hIL15-ABD, including the prolonged half-life and enhanced anticancer potency compared with WT IL15. Although, N-803 has a long half-life and IL15 activity enhanced by fused sushi-domain of IL15Ra, its production cost should be significant higher than that of hIL15-ABD, which is expressed by E. coli and refolded into active form robustly with relatively high yield. We added one paragraph in discussion section to address this issue (page 21).

They also do not mention the role of IL-15 in T resident memory cells which some regard as critical against cancers like melanoma.

Reply: Thanks for reviewer important comment. We have performed CD4+ T cells analysis based on our data; however, no obvious population induction was found in treatment group. Thus, we do not discuss the results that corrected to T memory in IL-15-ABD combined with anti-PD-L1.

All the comments had been revised point-by-point as showed in revised text file and marked as red color. Furthermore, we also confirmed our manuscript had followed publish criteria of Cancers. The relative revised version of manuscript body text was re-uploaded online. Thanks again for your kindly suggestion and consideration.

Sincerely,

Keng-Li Lan, M.D, Ph.D

Associate Professor, The Institute of Traditional Medicine, School of Medicine, National Yang Ming Chiao Tung University, Taipei, Taiwan.

Tel: +886-2-2826-7000 #7121, Email: kllan@ym.edu.tw

Reviewer 2 Report

The study of Dr Fei-Ting Hsu et al aims to evaluate the pharmacological properties and the antitumor activity of recombinant human IL15 protein fused to the albumin binding domain (hIL15-ABD) compared with wild-type IL15. The Authors demonstrates that chimeric IL15 recombinant molecule exhibits a much longer half-life in serum compared with IL15 after intraperitoneal injection in mice, and proved to exert increased antitumor activity on the colon cancer CT26 cells subcutaneously grafted in Balb/c mice. Furthermore, hIL15-ABD potentiates the anti-PD-L1 antitumor therapy, as evaluated with the colon cancer CT26 cells and the B16F10 melanoma cell line inoculated in Balb/c and C57BL6/J mice. This effect seems to involve an increased recruitment of CD8+ T cells and NK cells.

This study is of interest considering the limited efficacy of anti-PDL1 immunotherapy in some patients and the requirement to improve this treatment to convert refractory patients into responders

Minor comments

The title should specify that this is a preclinical study, e.g. "Recombinant human IL15 protein with albumin binding domain enhances efficacy of PD-L1 antibody via potentiating anti-tumor immunity in experimental mouse models of colon cancer and melanoma” or “Preclinical evaluation of recombinant human IL15 protein fused with albumin binding domain on anti-PDL1 immunotherapy efficiency and anti-tumor immunity in colon cancer and melanoma”…

Why the Authors did not use equimolar doses of hIL15 and hIL15-ABD to evaluate in vivo toxicity and tumor growth (Figure 3).

Lines 429-430, the Authors mentioned that they checked the VEGF level to demonstrate the decreasing inflammation after combination therapy. This assertion is not very clear for me. Combined therapy triggers inflammation. Especially, it seems that at least for B16F10 tumors, there is an increased endogenous IL15 level in mouse serum after hIL15-ABD and anti-PDL1 combined treatment. Is VEGF an effector or a biomarker of inflammation ?

Statistical analysis (lines 217-222) For comparison of more than two groups, ANNOVA test should be used as mentioned in the Material & Methods Section, and followed by a post-hoc test (e.g. Dunnett’s test for a comparison with control group, Tukey’s test to compare all groups). It is not clear which test in this study was performed following ANOVA.

The number of mice used in each experiment should be indicated

In many figures, the text is unreadable (Figs 3-5, 7). Please increase the size of the police.

Please, provide a size bar for the immunohistochemistry micrographs

As mentioned by the Authors, several other IL15 superagonists, including N-803 have been developed, some of them being under clinical trial. The Authors could discuss how hIL15-ABD compares with these analogs.

The panel L of Figure 7 could be used as a graphical abstract.

The Authors should check carefully de manuscript for misprints, e.g.

Line 30: change “In conclude” to “In conclusion” or “To conclude”,

Line 94: change “STA5” to “STAT5”,

The CT26 cell line originate from a colonic tumor chemo-induced by NNMU, please remove “rectal” lines 34, 97, 147, …

Line 159 “jetPEITM”, please put “TM” in superscript

Line 193 change “will” to “were”

Line 595 change “was incubated” to “were incubated”

Please note that during the edition process, all the Greek special characters (ug, INF-g, …) have been withdrawn.

Author Response

Cancers

RESVISED RESPONSES TO REVIEWERS

Manuscript Number: cancers-1111292

Manuscript Title: Preclinical evaluation of recombinant human IL15 protein fused with albumin binding domain on anti-PDL1 immunotherapy efficiency and anti-tumor immunity in colon cancer and melanoma

Authors: Fei-Ting Hsu, Yu-Chang Liu, Chang Liang Tsai, Po-Fu Yueh, Chih-Hsien Chang, Keng-Li Lan

Dear Reviewer

First of all, we would like to thanks for your valuable advices. We revised the main text according to your valuable comments, and all the revised words and sentences were marked as red color. Original comments from reviewer were written in black color, and the revised information was written as blue color as follow.

Major points:

The study of Dr Fei-Ting Hsu et al aims to evaluate the pharmacological properties and the antitumor activity of recombinant human IL15 protein fused to the albumin binding domain (hIL15-ABD) compared with wild-type IL15. The Authors demonstrates that chimeric IL15 recombinant molecule exhibits a much longer half-life in serum compared with IL15 after intraperitoneal injection in mice, and proved to exert increased antitumor activity on the colon cancer CT26 cells subcutaneously grafted in Balb/c mice. Furthermore, hIL15-ABD potentiates the anti-PD-L1 antitumor therapy, as evaluated with the colon cancer CT26 cells and the B16F10 melanoma cell line inoculated in Balb/c and C57BL6/J mice. This effect seems to involve an increased recruitment of CD8+ T cells and NK cells.

This study is of interest considering the limited efficacy of anti-PDL1 immunotherapy in some patients and the requirement to improve this treatment to convert refractory patients into responders

 Reply: Thanks for reviewer valuable comment.

Minor comments

The title should specify that this is a preclinical study, e.g. "Recombinant human IL15 protein with albumin binding domain enhances efficacy of PD-L1 antibody via potentiating anti-tumor immunity in experimental mouse models of colon cancer and melanoma” or “Preclinical evaluation of recombinant human IL15 protein fused with albumin binding domain on anti-PDL1 immunotherapy efficiency and anti-tumor immunity in colon cancer and melanoma”…

Reply: Thanks for reviewer valuable comment. We revised our tile in revised version.

Why the Authors did not use equimolar doses of hIL15 and hIL15-ABD to evaluate in vivo toxicity and tumor growth (Figure 3).

Reply: Thanks for reviewer valuable comment. Indeed, it could have been more appropriate to use just equimolar amount of proteins. We estimated that hIL15-ABD would have much impact on tumor bearing mice given its greatly increased half-life and AUC compared to those of IL15, therefore, we choose to use equal weight of IL-15 and hIL15-ABD of 5 mg for each injection, which represent 0.36 and 0.24 nanomole of hIL15 and hIL15-ABD, respectively.  We also added following information in manuscript “In light of the superior pharmacokinetic profiles of hIL15-ABD and to demonstrate the potent in vivo anticancer effect of hIL15-ABD, we used 5 mg for each injection, which represent 0.36 and 0.24 nanomole of hIL15 and hIL15-ABD, respectively, instead of using equimolar proteins. ” We added this paragraph in page 9.

Lines 429-430, the Authors mentioned that they checked the VEGF level to demonstrate the decreasing inflammation after combination therapy. This assertion is not very clear for me. Combined therapy triggers inflammation. Especially, it seems that at least for B16F10 tumors, there is an increased endogenous IL15 level in mouse serum after hIL15-ABD and anti-PDL1 combined treatment. Is VEGF an effector or a biomarker of inflammation?

Reply: Thanks for reviewer important comment. We revised this sentences as “VEGF (Figure 6I) secretion level in mouse serum to demonstrate the decreasing of immunosuppressive factor after combination therapy”. VEGF was known to play role on immune regulation and thus enhancing the function of immune suppressive cells. Therefore, it has been recognized as immunosuppressive factor.

Statistical analysis (lines 217-222) For comparison of more than two groups, ANNOVA test should be used as mentioned in the Material & Methods Section, and followed by a post-hoc test (e.g. Dunnett’s test for a comparison with control group, Tukey’s test to compare all groups). It is not clear which test in this study was performed following ANOVA.

Reply: Thanks for reviewer important comment. In this study, Tukey’s test was used to compare all groups as post-hoc test.

The number of mice used in each experiment should be indicated

Reply: Thanks for reviewer valuable comment. All experiment was repeated at least twice (n=6). We added this information into material and methods.

In many figures, the text is unreadable (Figs 3-5, 7). Please increase the size of the police.

Reply: Thanks for reviewer valuable comment. We revised figure 3-7 and enlarge the labeling within figure.

Please, provide a size bar for the immunohistochemistry micrographs

Reply: Thanks for reviewer valuable comment. We have added scale bar information in each relative figure legend.

As mentioned by the Authors, several other IL15 superagonists, including N-803 have been developed, some of them being under clinical trial. The Authors could discuss how hIL15-ABD compares with these analogs.

Reply: In addition to the two IL15 related therapeutics, WT IL15 and N-803, currently in clinical trials, we added more information on two PD-L1 targeting IL15, KD033 and N-809 in the manuscript. Additionally, we outlined the potential advantages of hIL15-ABD, including the prolonged half-life and enhanced anticancer potency compared with WT IL15. Although, N-803 has a long half-life and IL15 activity enhanced by fused sushi-domain of IL15Ra, its production cost should be significant higher than that of hIL15-ABD, which is expressed by E. coli and refolded into active form robustly with relatively high yield. We added one paragraph in discussion section to address this issue (page 21).

The panel L of Figure 7 could be used as a graphical abstract.

Reply: Thanks for reviewer valuable comment. We revised it and provide it as graphic abstract.

The Authors should check carefully de manuscript for misprints, e.g.

Line 30: change “In conclude” to “In conclusion” or “To conclude”,

Reply: Thanks for reviewer valuable comment. We revised it.

Line 94: change “STA5” to “STAT5”,

Reply: Thanks for reviewer valuable comment. We revised it.

The CT26 cell line originate from a colonic tumor chemo-induced by NNMU, please remove “rectal” lines 34, 97, 147, …

Reply: Thanks for reviewer valuable comment. We revised all the description of colon rectal cancer into colon cancer.

Line 159 “jetPEITM”, please put “TM” in superscript

Reply: Thanks for reviewer valuable comment. We revised it.

Line 193 change “will” to “were”

Reply: Thanks for reviewer comment. We revised this part as requested.

Line 595 change “was incubated” to “were incubated”…

Reply: Thanks for reviewer comment. We revised this part as requested.

All the comments had been revised point-by-point as showed in revised text file and marked as red color. Furthermore, we also confirmed our manuscript had followed publish criteria of Cancers. The relative revised version of manuscript body text was re-uploaded online. Thanks again for your kindly suggestion and consideration.

Sincerely,

Keng-Li Lan, M.D, Ph.D

Associate Professor, The Institute of Traditional Medicine, School of Medicine, National Yang Ming Chiao Tung University, Taipei, Taiwan.

Tel: +886-2-2826-7000 #7121, Email: kllan@ym.edu.tw